# Combined Treatment of Cancer Cells Using Allyl Palladium Complexes Bearing Purine-Based NHC Ligands and Molecules Targeting MicroRNAs miR-221-3p and miR-222-3p: Synergistic Effects on Apoptosis

**DOI:** 10.3390/pharmaceutics15051332

**Published:** 2023-04-24

**Authors:** Chiara Tupini, Matteo Zurlo, Jessica Gasparello, Irene Lodi, Alessia Finotti, Thomas Scattolin, Fabiano Visentin, Roberto Gambari, Ilaria Lampronti

**Affiliations:** 1Department of Life Sciences and Biotechnology, University of Ferrara, 44121 Ferrara, Italy; 2Center of Innovative Therapies for Cystic Fibrosis (InnThera4CF), University of Ferrara, 44121 Ferrara, Italy; 3Dipartimento di Scienze Chimiche, Università degli Studi di Padova, 35131 Padova, Italy; 4Dipartimento di Scienze Molecolari e Nanosistemi, University Ca’ Foscari, 30174 Venezia-Mestre, Italy

**Keywords:** microRNA, anti-miR, antagomiRNAs, miR-221, miR-222, glioblastoma, colon cancer, apoptosis, combination therapy, organopalladium complexes, metal-based drugs

## Abstract

Combined treatments employing lower concentrations of different drugs are used and studied to develop new and more effective anticancer therapeutic approaches. The combination therapy could be of great interest in the controlling of cancer. Regarding this, our research group has recently shown that peptide nucleic acids (PNAs) that target miR-221 are very effective and functional in inducing apoptosis of many tumor cells, including glioblastoma and colon cancer cells. Moreover, in a recent paper, we described a series of new palladium allyl complexes showing a strong antiproliferative activity on different tumor cell lines. The present study was aimed to analyze and validate the biological effects of the most active compounds tested, in combination with antagomiRNA molecules targeting two miRNAs, miR-221-3p and miR-222-3p. The obtained results show that a “combination therapy”, produced by combining the antagomiRNAs targeting miR-221-3p, miR-222-3p and the palladium allyl complex **4d**, is very effective in inducing apoptosis, supporting the concept that the combination treatment of cancer cells with antagomiRNAs targeting a specific upregulated oncomiRNAs (in this study miR-221-3p and miR-222-3p) and metal-based compounds represents a promising therapeutic strategy to increase the efficacy of the antitumor protocol, reducing side effects at the same time.

## 1. Introduction

Combination treatments (“combo-therapy”) seem to have a high potential for the development of novel anticancer therapeutic approaches [1,2,3,4,5]. The primary goal of combined treatments is based on the concept and prospect of employing two or more agents in combination in order to have the same biological or therapeutic effect while using lower dosages to reduce recognized adverse effects of the employed agents [1].

In addition, studies have confirmed that an antineoplastic combined therapy can reduce possible acquired resistance [1,2,5]. Therefore, the overall interest in anticancer combined therapy is currently very high [3].

Concerning combined therapeutic approaches, we have recently demonstrated and reported that antitumor derivatives can be combined with molecules targeting specific microRNAs [6,7,8,9], short noncoding RNAs that can operate as gene regulators by directly inhibiting translation or triggering the cleavage of target mRNA transcripts [10,11,12]. MiRNAs may play a role in the pathogenesis of cancer, according to well-established and acknowledged data [13,14]. OncomiRNAs and metastamiRNAs are the terms used to describe miRNAs that are increased in cancer and subsequently cause the downregulation of target tumor suppressor mRNAs [15].

Several studies are available demonstrating synergistic effects of miRNA targeting and use of anticancer drugs [16,17,18]. For instance, Sun et al. developed a core–shell supramolecular nanovector of “chitosome” loaded with Docetaxel (a well-known antineoplastic agent) and with an anti-miRNA molecule against miR-21 to improve chemoresistance in breast tumor [4]. Interestingly, the combination of a conventional chemotherapeutic agent together with the suppression of a specific upregulated oncomiRNA makes cancer cells more sensitive to the treatment, in this case a proposed mechanism is the upregulation of PTEN (an important oncosuppressor that is often inactivated in different types of cancer) which is targeted from miR-21.

In this study, we have employed antagomiRNA molecules targeting miR-221-3p and miR-222-3p on colorectal cancer (CRC) and glioblastoma (GBM) cell lines. MiR-221-3p and miR-222-3p have been selected considering their overexpression in patients with CRC and GBM [19,20,21,22,23]. Moreover, miR-222-3p-targeting decreases cell migration and metastasis using an in vivo GBM [24] or a CRC experimental model systems [25]. Furthermore, the oncogenic role of miR-221-3p was also reported in other types of tumors, such as liver, pancreatic and lung cancer [26,27,28].

Regarding our research activity in this field of investigation, we have designed, synthesized and tested new palladium allyl complexes bearing purine-based *N*-heterocyclic carbenes derived from the caffeine, theophylline and theobromine scaffolds, characterized by evident cytotoxic and proapoptotic effects on the A2780, Cisplatin-sensitive, and SKOV-3, Cisplatin-resistant, ovarian cancer cell lines [29]. The accurate choice of the supporting ligands is essential for realizing the expected results, and the palladium-allyl fragment was found capable of inducing general cytotoxic effects. The conclusion of this previously published study is that these compounds, and other well-defined organopalladium derivatives, are of great interest for the study of novel anticancer strategies and treatments [29,30,31,32].

The first goal of the present research project was to validate the activity of three different palladium allyl derivatives (**4d**, **5d** and **7a**) on cell proliferation of HT29, Ls-174-T and LoVo colon cancer cells, or U251 and T98G glioblastoma cell lines. These three palladium complexes, bearing purine-based carbenes, were designed and synthesized by our research group and previously tested on human ovarian cancer A2780 and SKOV-3 cells [29].

The second aim of our study was to expand the analysis by deepening the activity of these derivatives in order to identify the most bioactive compound, i.e., the best candidate to be used in combination with antagomiRNA molecules against miR-221-3p and miR-222-3p on colon cancer and glioblastoma cancer cells. The selected final endpoint was the ability to induce apoptosis.

## 2. Materials and Methods

### 2.1. Chemistry and Reagents

The research group of Prof. Visentin synthesized the palladium compounds **4d**, **5d** and **7a** (Figure 1A); the exploited synthetic procedure has been already reported [29]. All cell cultures required the use of RPMI 1640 medium (cat.no. BE12-702F, Lonza Biosciences, Basel, Switzerland) supplemented with 10% FBS (cat.no. S1400, Biowest, Nuaillè, France) and 100 mg/mL streptomycin and 100 IU/mL penicillin (cat.no. 11074440001, Sigma-Aldrich Merck KGaA). Trypsin-EDTA (cat.no. 59428C, Sigma-Aldrich Merck KGaA, Darmstadt, Hesse, Germany) was utilized to detach the adherent cells, and the Muse^®^ Annexin-V & Dead Cell kit (cat.no. MCH100105, Luminex Corporation, Austin, TX, USA), Muse^®^ Caspase-3/7 kit (cat.no. MCH100108, Luminex Corporation, Austin, TX, USA) was used to perform flow cytometry assays.

### 2.2. Cell Growth Condition and Cell Lines

Human colon cancer HT29 [33], Ls-174-T [34] and LoVo [35], glioblastoma U251 [36] and T98G [37] cell lines were employed. Cells were seeded in a 12-well plate and maintained at standard conditions (37 °C in a humidified 5% CO_2_ atmosphere) for 24 h, then treated and incubated for additional 48 h. Following 48 h of treatment, cells were detached from plates with trypsin and counted using a BECKMAN COULTER Z2 cell counter (Beckman, Pasadena, CA, USA). Then, the IC_50_ values for each compound were calculated. The IC_50_ represents the 50% inhibitory concentration, which is defined as the compound concentration inhibiting cell proliferation of 50% [7]. The reported IC_50_ values are average values (±standard deviation) relative to three independent experiments.

### 2.3. Combined Treatment and Cell Transfection

Cells were seeded in 12-well plates and maintained 24 h at standard conditions (37 °C in a humidified atmosphere 5% CO_2_) prior to proceeding with the combined treatment. The day after plating, cells were treated and incubated 48 h before proceeding with Apoptosis measurement and RNA extraction; CisPt and compound **4d** were employed at sub-optimal concentration (lower than IC_50_ tested on each cell line). For antimiRNAs transfection, Lipofectamine RNAiMAX (cat.no. 13778075, ThermoFisher, Waltham, MA, USA) was employed following manufacturers’ protocol. In this case, 200 nM antimiR-221-3pp or antimiR-222-3p was diluted in 50 μL of Opti-MEM medium, mixed and incubated 5 min at room temperature with Lipofectamine RNAiMAX in diluted in 50 μL of Opti-MEM prior to treat cells. Cells were treated with CisPt, compound **4d**, antimiR-221 or 222 singularly and in combination (CisPt+antimiR-221 or 222 and compound **4d**+antimiR-221 or 222); in addition, we put the same amount of Opi-MEM medium in each well and included the proper vehicle controls for single and combined treatment (EtOH, Lipofectamine, or both).

### 2.4. RNA Extraction

After 48 h of treatment, cells were gently detached with Trypsin-EDTA (cat.no. 59428C, Sigma-Aldrich Merck KGaA, Darmstadt, Hesse, Germany), collected, pelleted in a 1.5 mL tube, and finally lysed with 500 μL of Tri-Reagent (cat.no. 93289, Sigma-Aldrich Merck, Darmstadt, Hesse, Germany). Following a cold 75% ethanol wash, the isolated RNA was kept at −80 °C. Before usage, obtained RNA was vacuum-dried and dissolved in pure and nuclease-free water [9].

### 2.5. Quantification of miRNAs

MicroRNAs cellular content was investigated using TaqMan MicroRNA Reverse Transcription Kit (cat.no. 43-665-96, Life Technologies, Carlsbad, CA, USA) with RT-qPCR primers and probes that are specific for each target miRNAs tested (listed in Table 1 and purchased from Applied Biosystems, ThermoFisher, Waltham, MA, USA). Each sample was run in duplicate employing TaqMan Universal PCR Master Mix, no AmpErase UNG 2X (cat.no 4324018- ThermoFisher, Waltham, MA, USA) and the CFX96 Touch Real-Time PCR Detection System (BioRad, Hercules, CA, USA). The PCR reaction protocol utilized was the following: 95 °C for 10 min, 95 °C for 15 s, 60 °C for 1 min (last two steps repeated for 50 cycles). Data were obtained and analyzed using Bio-Rad CFX Manager Software (Bio-Rad, Hercules, CA, USA). Relative gene expression was calculated using 2^−ΔΔCt^ method and data normalization was performed using snRNA U6 and hsa-let-7c as endogenous control [9].

### 2.6. Analysis of Apoptosis

Apoptosis levels were assayed with Guava^®^ Muse^®^ Cell Analyzer instrument and kits (Luminex Corporation, Austin, TX, USA). After 48h of treatment, cell cultures were washed twice with PBS, detached by trypsinization and resuspended in RPMI complete medium. Lastly, 100 μL of cell suspension was incubated for 20 min, protected from light, at room temperature with 100 μL of Muse^®^ Annexin V & 7-AAD (7-aminoactinomycin D) Dead Cell reagent. Cells in the early stage of the apoptotic process (represented in lower right quadrants in the plots) were stained with the Annexin V-PE conjugated; cells in the late phase of apoptosis (upper right quadrants) were stained with Annexin V-PE plus 7-AAD; the necrotic cells were stained only by 7-AAD (upper left quadrants) while live cells showed no staining (lower left quadrants). Following that, samples were obtained using a Guava^®^ Muse^®^ Cell Analyzer (Luminex Corporation, Austin, TX, USA), and data was analyzed using the Luminex Annexin V and Dead Cell Software Module (as previously reported [8]).

Similarly, caspase-3/7 staining was performed with Guava^®^ Muse^®^ Cell Analyzer Caspase-3/7 kit (Luminex Corporation, Austin, TX, USA) following the same protocol of treatment and detachment explained above and following manufacturer’s instruction.

### 2.7. Statistics

All the reported results are expressed as mean ± standard deviation. The Prism Software was utilized to calculate the comparisons between groups using two-tail paired *t*-test or ANOVA followed by Dunnet’s multiple comparison. Statistical significance was defined as follows: *p* < 0.05 (*, significant), *p* < 0.01 (**, highly significant), *p* < 0.001 (***, highly significant).

## 3. Results

### 3.1. Structures of the Molecules Employed in This Study and Their Effects on Cell Proliferation

Figure 1A shows the chemical structure of the allyl palladium complexes **4d**, **5d** and **7a**. These compounds have been selected from a panel of palladium allyl complexes synthetized and characterized by some of us in a previously published study [29], which had already shown an excellent antiproliferative activity toward two different lines of ovarian cancer cells. The choice of one phosphine and one N-heterocyclic carbene as ancillary ligands has been made considering the ability of these ligands to stabilize the complexes of transition metals, especially those of 2nd and 3rd transition series. Moreover, the purine-based NHCs should ensure better compatibility with biological environment.

Figure 1B shows the effects of compounds **4d**, **5d** and **7a** on cell proliferation of colon cancer (HT29, Ls-174-T and LoVo) and glioblastoma (U251 and T98G) cell lines. After 72 h of treatment, cell cultures were analyzed to verify the antiproliferative effects of the allyl palladium complexes **4d**, **5d** and **7a**, in comparison with the Cisplatin (CisPt) used at different concentrations. The obtained data indicate that the most effective inhibition of in vitro cell growth is achieved when **4d** is used. Figure 1C describes the effects of CisPt on the same cell lines. CisPt was used also in our previous studies and chosen as a positive control, because it is a well-known complex capable of causing apoptosis in various tumor cell lines and, for this reason, still utilized as antineoplastic drug to treat aggressive ovarian and orogastric cancers, although it has many known side effects.

Considering the highest effects of compound **4d**, this derivative was selected for the study.

The observed antiproliferative activity of **4d** was demonstrated to be associated with activation of apoptosis when the Annexin V assay was employed (Figure 2). Interestingly, induction of apoptosis was detected with high efficiency when compound **4d** was assayed using HT29 (Figure 2A), Ls-174-T (Figure 2B), LoVo (Figure 2C), U251 (Figure 2D) and T98G (Figure 2E) cell lines. Moreover, the effects were remarkable when comparison was conducted with CisPt. For instance, when HT29 colon cancer cells were employed, **4d** was able to induce 51.60% of late apoptotic cells compared with the 26.05% of late apoptotic cells induced by CisPt (Figure 2A).

In order to verify whether (a) compound **4d** synergizes with molecules inhibiting antiapoptotic microRNAs and (b) combined treatments have appreciable effects in T98G drug resistant cells, oligonucleotides able to inhibit miR-221-3p and miR-222-3p were employed. These miRNA targets were selected considering their well-established roles as antiapoptotic regulators [38]. Moreover, the miR-221/miR-222-mRNA networks are well known. For instance, among the potential apoptotic-associated mRNAs, ATG10, CDKN1B/p27, BMF, APAF-1, PTEN, p27(kip1), p57(kip2) and PUMA have been validated as miR-221/miR-222 targets [39,40,41,42,43,44,45,46]. Appendix A section, shows the interactions between miR-221/miR-222 miRNAs and the PUMA 3′-UTR mRNA.

### 3.2. Compound ***4d*** Does Not Affect Expression of miR-221-3p and miR-222-3p

Compound **4d** does not affect miR-221-3p and miR-222-3p in treated HT29 and U251 cells. This experiment, shown in Figure 3, was conducted by exposing, for 48 h, human colon cancer HT29 and human glioblastoma U251 cells to compound **4d** or CisPt at IC_50_ concentration. After this period of cell culture, RNA was isolated, and miR-221-3p (Figure 3A) and miR-222-3p (Figure 3B) were quantified by RT-qPCR. The conclusion of this experiment is that compound **4d** does not cause inhibition of the expression of both miR-221-3p and miR-222-3p; similar results were obtained using the T98G temozolomide-resistant glioblastoma cell line.

### 3.3. Co-Treatment of Colon Cancer HT29 and Glioblastoma U251 Cells with ***4d*** and miR-221-3p and miR-222-3p Inhibitors: Effects on Apoptosis Induction

Representative experiments were focused on the effects of single and combined treatments on the induction of apoptosis of HT29 (Figure 4) and U251 (Figure 5) cells. We have comparatively tested the induction of apoptosis (performing Annexin V assay) in the presence of (a) the employed vectors or delivery systems (EtOH, lipofectamine, EtOH+ lipofectamine), (b) the singularly administered **4d**, miR-221-3p and miR-222-3p inhibitors and (c) the combined treatments (4d + miR-221-3p inhibitor and **4d** + miR-222-3p inhibitor).

The representative experiments shown in Figure 4 and Figure 5 and summarized in Figure 6 conclusively show that the employed vectors or delivery systems (EtOH, lipofectamine, EtOH + lipofectamine) do not induce a significant increase in apoptosis. A second observation is that the singular administration of low concentrations of **4d**, CisPt, miR-221-3p inhibitor and miR-222-3p inhibitor stimulates apoptosis. However, the most important conclusion is that the combined treatments based on compound **4d** (**4d** + miR-221-3p inhibitor and **4d** + miR-222-3p inhibitor) and on CisPt (CisPt + miR-221-3p inhibitor and CisPt + miR-222-3p inhibitor) lead to an increase in apoptotic cells percentage that was found to be always higher than the sum of the singular treatments. For instance, the sum of the treatment with compound **4d** and miR-221-3p inhibitor was 24.11%, while the combined **4d** + miR-221 inhibitor was 46.50% (Figure 6A). This very interesting and reproducible result demonstrates a synergism of action. In addition, the combined treatments based on CisPt demonstrated synergistic effects on induction of apoptosis. As expected, down-regulation of miR-221-3p and miR-222-3p relative content was observed in all the treatments in which the miR-221-3p and miR-222-3p inhibitors were used (Figure 6B,D,F,H).

### 3.4. Cell Apoptosis Study on Glioblastoma Temozolomide-Resistant T98G Cells

To verify whether the synergism of action between compound **4d** and the miR-221-3p and miR-222-3p inhibitors occurs also in the temozolomide-resistant T98G cells, the protocol employed on U251 and HT29 cells and described in Figure 4 and Figure 5 was used. The results are shown in Figure 7 and demonstrate that a synergistic action between compound **4d** and the miR-221-3p inhibitor is operating also on T98G cells.

The combined treatment, graphed in Figure 8, leads to a clear increase in the apoptosis process which was observed to be higher than the simple sum of the effects obtained using single administration of compound **4d** and the miR-221 inhibitor (indicated by the dashed lines). On the contrary no synergistic effects were found with combined treatments based on the miR-222-3p inhibitor and on CisPt.

### 3.5. Co-Treatment of Glioblastoma U251 Cells with ***4d*** and miR-221-3p and miR-222-3p Inhibitors: Effects on Caspase-3/7 Activation

The effectors in the enzymatic cascade able to activate the apoptotic process are caspases 3 and 7. They are considered biomarkers because they show different phases of the programmed cell death process. To further prove apoptotic protein activation following treatment with compound **4d** and miR-221/222 inhibitors, we examined Caspase-3/7 activation by flow cytometry technique as explained in the methods section in U251 glioblastoma cell line. A representative experiment is shown in Figure 9A, while in Figure 9B total apoptotic/dead cells are shown as mean ± standard deviation (n = 3).

While compound **4d** and antimiR-221/222 singularly did not show a significant increase in caspase-3/7 activation, the combined treatment with compound **4d** and antimiR-221 or antimiR-222 again showed synergistic effects, resulting in a marked increase in caspase-3/7 activation, suggesting that programmed cell death is actively induced following combination treatment.

## 4. Discussion

Colon cancer (CRC) and Glioblastoma (GBM) patients express high levels of onco-miRNAs, such as miR-221-3p and miR-222-3p, both exerting antiapoptotic effects and promoting malignant progression [21,22,23,24]. The involvement of miR-221-3p and miR-222-3p in CRC has been reported in the studies published by Sanhong et al. [47], demonstrating higher levels of miR-221 and miR-222 in human CRC tissues in respect to healthy colon tissues; this was found to be associated with increased expression of RelA and STAT3 mRNAs; accordingly, interference with the biological activity of miR-221 and miR-222 reduces in vivo growth of colon cancers [47]. The research published by Xu et al. [24] demonstrated that inhibition of both miR-221 and miR-222 significantly lowers proliferation, invasion, migration and angiogenesis of glioblastoma cells in vitro and in vivo. This finding supports the involvement of miR-221-3p in GBM. p-JAK2/JAK2 and p-STAT3/STAT3 pathway activation, levels of various matrix metalloproteinases (including MMP-2 and MMP-9) and levels of vascular endothelial growth factor (VEGF) are all decreased when miR-221-3p and miR-222-3p are inhibited [24].

Considering all these data, we might suggest the miR-221/222 cluster as a pertinent therapeutic target in procedures intended to reduce the invasiveness of GBM and CRC cells during carcinogenesis. An interesting approach that employs combined treatment has been recently proposed by our group, associating compounds interfering with tubulin polymerization and specific antimiRNA molecules targeting upregulated oncomiRNAs (such as miR-221-3p or miR-10b-5p). Combined treatment appears to have synergistic effects on the activation of apoptosis and cell-cycle alterations, demonstrating that this therapeutical strategy can increase the efficacy of anticancer treatments [7,8].

Combined treatments are of clinical interest in the development of anticancer protocols [1,2,3,4,5]. This method limits adverse effects by allowing for the same biological or therapeutic effect to be achieved with two or more medicines administered at lower concentrations [1]. In light of this, combination therapy may be useful in the treatment of glioblastoma, a deadly malignant tumor that requires novel therapeutic choices. In fact, the first-line medicine now employed (temozolomide) is only able to slightly lengthen the life expectancy of GBM patients, who frequently develop chemotherapy resistance. Therefore, there is actually no effective pharmaceutical strategy for glioblastoma, and novel therapeutic protocols are highly needed for clinical validation [48,49,50].

Using inhibitors of miR-221-3p and miR-222-3p along with the palladium allyl derivative **4d**, one of the most potent compounds identified by Scattolin et al. [29], we described a novel “combo-therapy” in this study.

The combined treatments of HT29 and U251 cells based on compound **4d** (4d + miR-221-3p inhibitor and 4d + miR-222-3p inhibitor) and on CisPt (CisPt + miR-221-3p inhibitor and CisPt + miR-222-3p inhibitor) led to an apoptosis induction that was found to be always higher than the sum of the different treatments. The Combination Index (CI), based on the apoptosis values at different dosages of the drugs, was calculated using the Chou-Talalay method [51]; since the CI was always <1, these obtained results indicate synergistic (rather than additional) effects. As expected, inhibition of miR-221-3p and miR-222-3p was observed in all the treatments in which the miR-221-3p and miR-222-3p inhibitors were used.

Interestingly, when the temozolomide-resistant T98G cell line was used in the experiment, the combined therapy increased apoptosis, in this case more than the combined effects of the single administrations of compound **4d** and the miR-221 inhibitor, indicating a synergistic, rather than additive, effect.

In conclusion, the combined use of metal-based anticancer agent (compound **4d**) and upregulated “oncomiRNA” inhibitors (miR-221-3p and miR-222-3p in this work) to treat cancer cells is a promising method in the field of developing efficient anticancer therapies, according to our results.

## Figures and Tables

**Figure 1 pharmaceutics-15-01332-f001:**
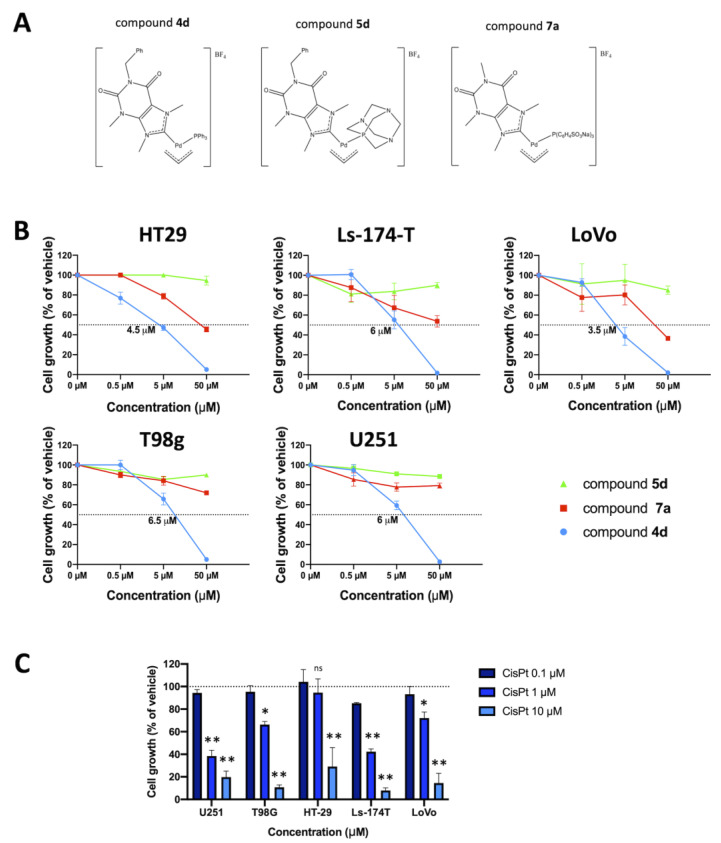
Chemical structure of molecules employed in this study and their effects on cell proliferation. (**A**) Chemical structure of the allyl palladium complexes **4d**, **5d** and **7a**. (**B**) Effects of compounds **4d** (blue lines and symbols), **5d** (green lines and symbols) and **7a** (red lines and symbols) on cell proliferation of different colon cancer (HT29, Ls-174-T and LoVo) and glioblastoma (U251 and T98G) cell lines, after 72 h of incubation; crossing of colored lines with the dotted line represents the IC_50_ value. (**C**) Effects of Cisplatin (CisPt) at different concentrations (0.1, 1, 10 μM) on cell proliferation of the same cell lines. Results represent mean ± SD (n = 3). *p* < 0.05 (*, significant), *p* < 0.01 (**; highly significant), ns = not significant, in comparison to the negative control (untreated cells), shown by the dotted line (100% cell growth).

**Figure 2 pharmaceutics-15-01332-f002:**
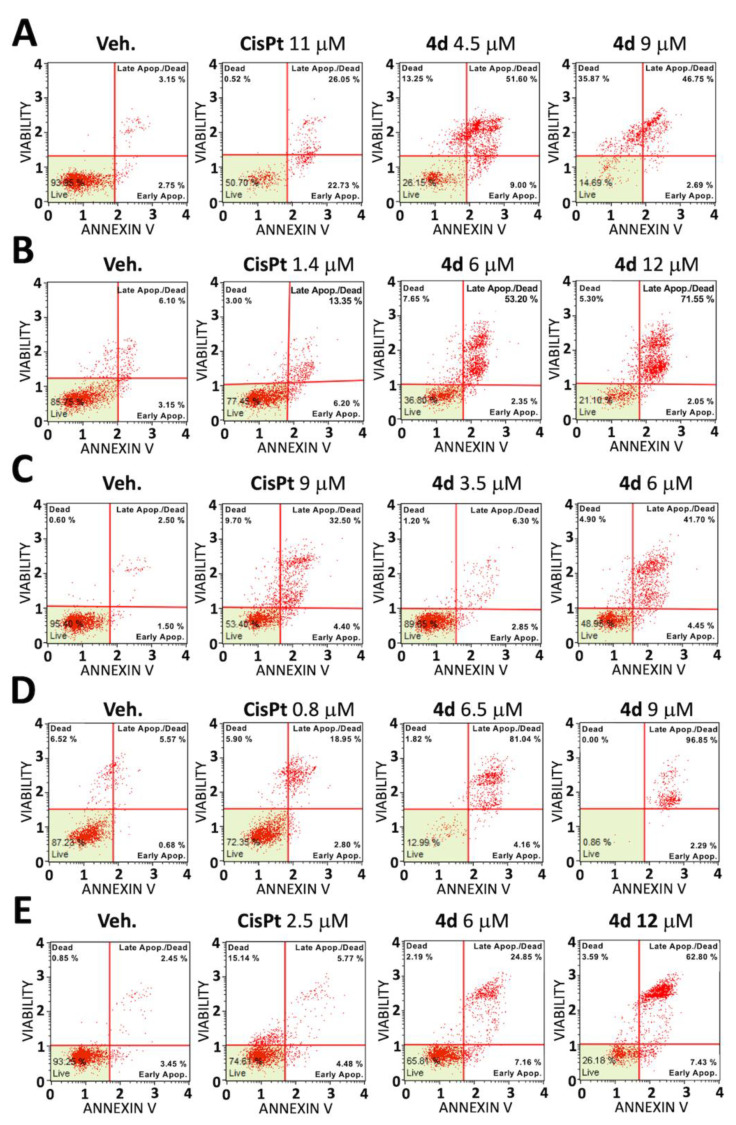
Induction of apoptosis of compound **4d** and Cisplatin (CisPt) in representative experiments. Annexin V/7-AAD assay was employed. HT29 (**A**), Ls-174-T (**B**), LoVo (**C**), U251 (**D**) and T98G (**E**) cells were treated with compound **4d** at IC_50_ and IC_75_ concentration or Cisplatin (IC_50_) for 48 h and then assayed for apoptosis induction, in comparison with cells treated with vehicle (veh.).

**Figure 3 pharmaceutics-15-01332-f003:**
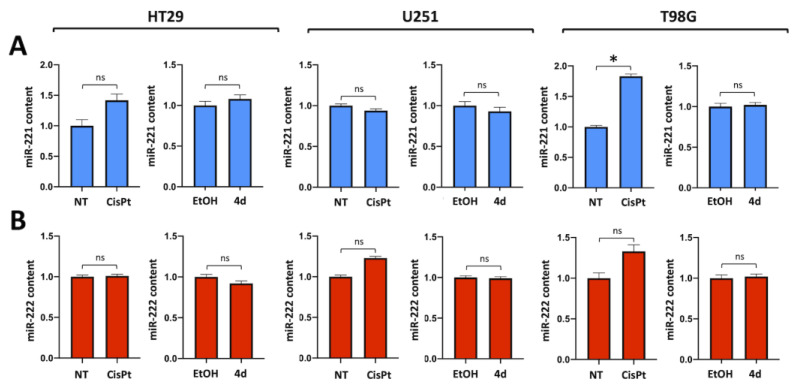
Cisplatin (CisPt) and compound **4d** do not act downregulating miR221 (**A**) or miR222 (**B**) in colorectal HT29 and glioblastomaU251 and T98G cell lines. CisPt and complex **4d** treatment normally do not interfere with miR-221/222 expression in tested cell lines. RT-qPCR data were normalized with housekeeping U6 snRNA, cells treated with water-soluble CisPt were compared with untreated cells (NT) while cells treated with **4d**, soluble in ethanol, were compared with cells treated with vehicle (EtOH). Results represent mean ± SD (n = 3). *p* < 0.05 (*, significant), ns = not significant.

**Figure 4 pharmaceutics-15-01332-f004:**
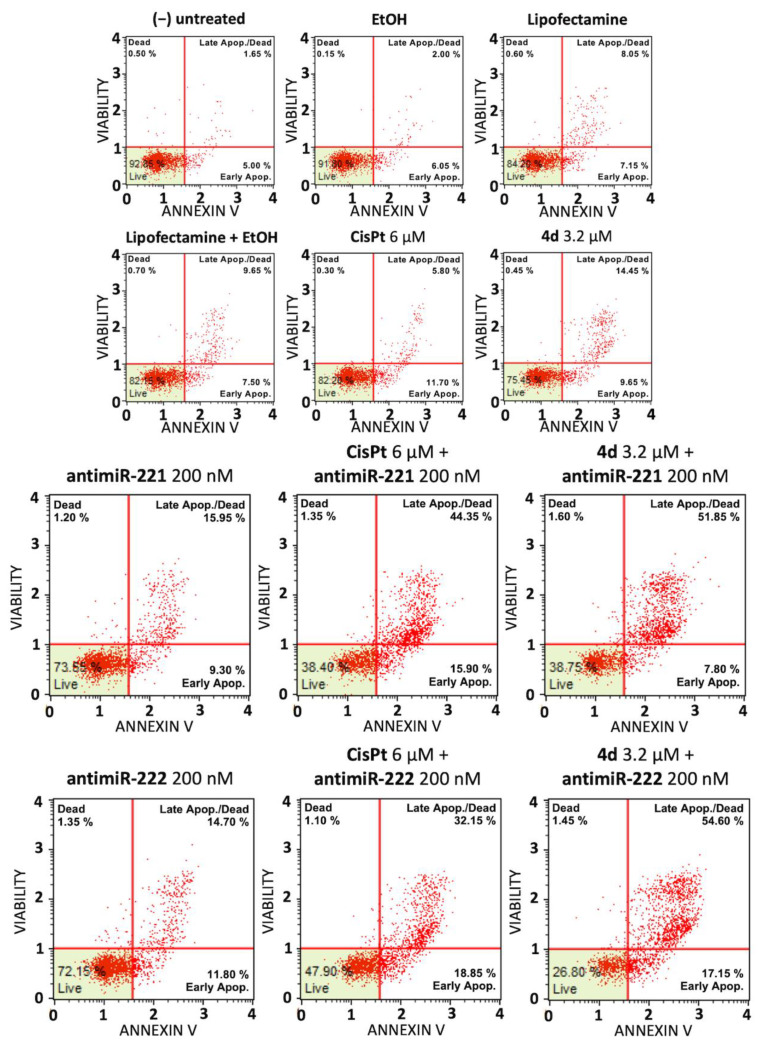
Induction of apoptosis of combined treatments based on the compound **4d** and inhibitors of miR-221-3p and miR-222-3p: colon cancer HT29 cell line. Annexin V assay was employed. HT29 cells were treated for 48 h with the indicated experimental conditions, i.e., with the employed vectors or delivery systems (EtOH, lipofectamine, EtOH + lipofectamine) and the singularly administered **4d** and the positive control CisPt (**upper** panel), with singularly administered miR-221-3p and miR-222-3p inhibitors and with the combined treatments using the miR-221-3p and miR-222-3p inhibitors and **4d** or CisPt (**lower** panel).

**Figure 5 pharmaceutics-15-01332-f005:**
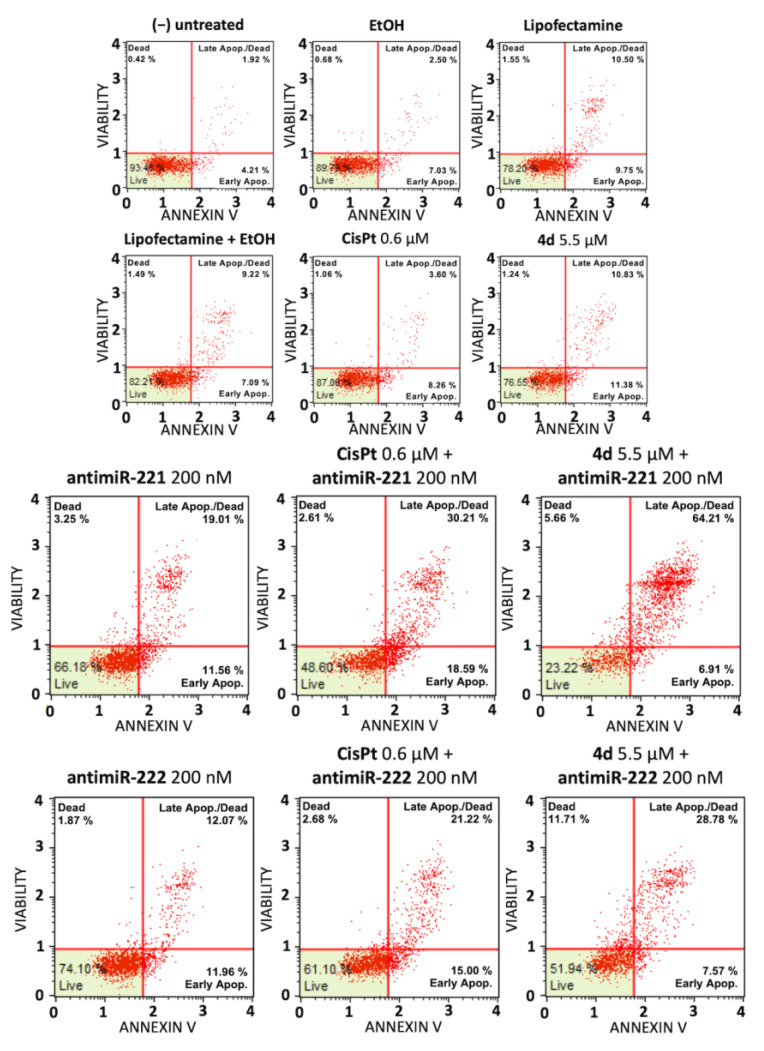
Induction of apoptosis of combined treatments based on the compound **4d** and inhibitors of miR-221-3p and miR-222-3p: glioblastoma U251 cell line. Annexin V assay was employed. U251 cells were treated for 48 h with the indicated experimental conditions, i.e., with the employed vectors or delivery systems (EtOH, lipofectamine, EtOH + lipofectamine) and with the singularly administered **4d** and the positive control CisPt (**upper** panel), with singularly administered miR-221-3p and miR-222-3p inhibitors and with the combined treatments using the miR-221-3p and miR-222-3p inhibitors and **4d** or CisPt (**lower** panel).

**Figure 6 pharmaceutics-15-01332-f006:**
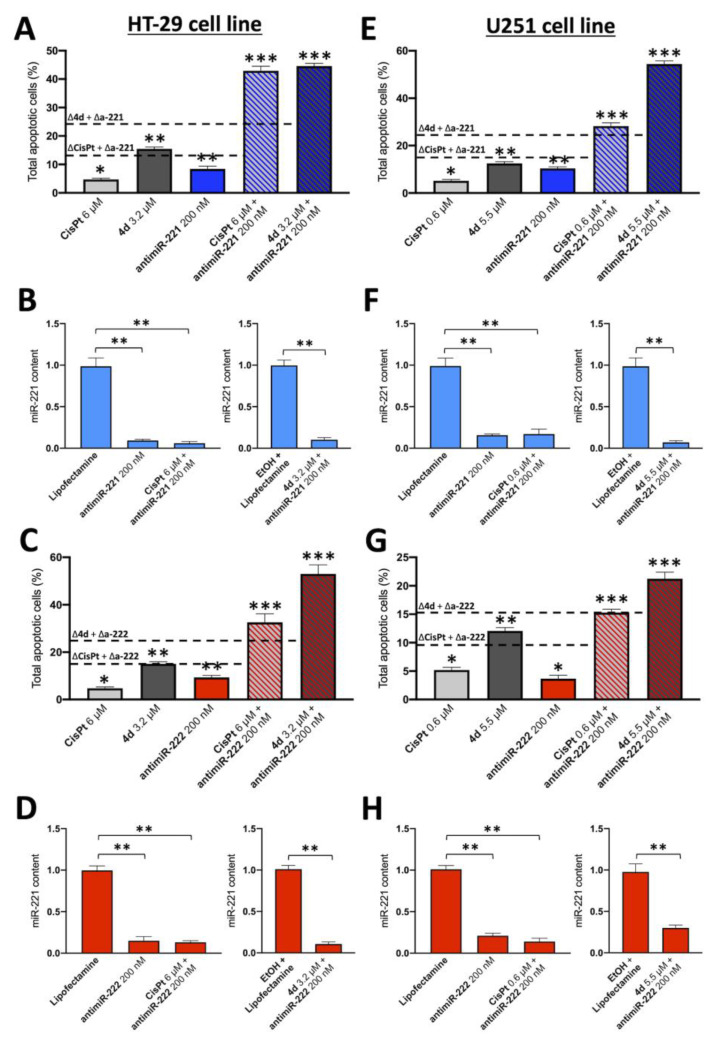
Summary of the experiments performed on the induction of apoptosis of combined treatments based on the compound **4d** and inhibitors of miR-221-3p and miR-222-3p. Experiments performed on HT29 cell line (**A**–**D**) are displayed on the left, while U251 cells (**E**–**H**) are displayed on the right; (**A**,**C**,**E**,**G**): effects on the induction of apoptosis in combined treatments based on miR-221-3p (**A**,**E**) and miR-222-3p (**C**,**G**) inhibitors; (**B**,**D**,**F**,**H**): effects of the indicated treatments on accumulation of miR-221-3p (**B**,**F**) and miR-222-3p (**D**,**H**). The dotted lines of panels (**A**,**C**,**E**,**G**) represent the sum of the singularly administered compounds, as indicated. Results represent mean ± SD (n = 3). *p* < 0.05 (*, significant), *p* < 0.01 (**; highly significant), *p* < 0.001 (***; highly significant).

**Figure 7 pharmaceutics-15-01332-f007:**
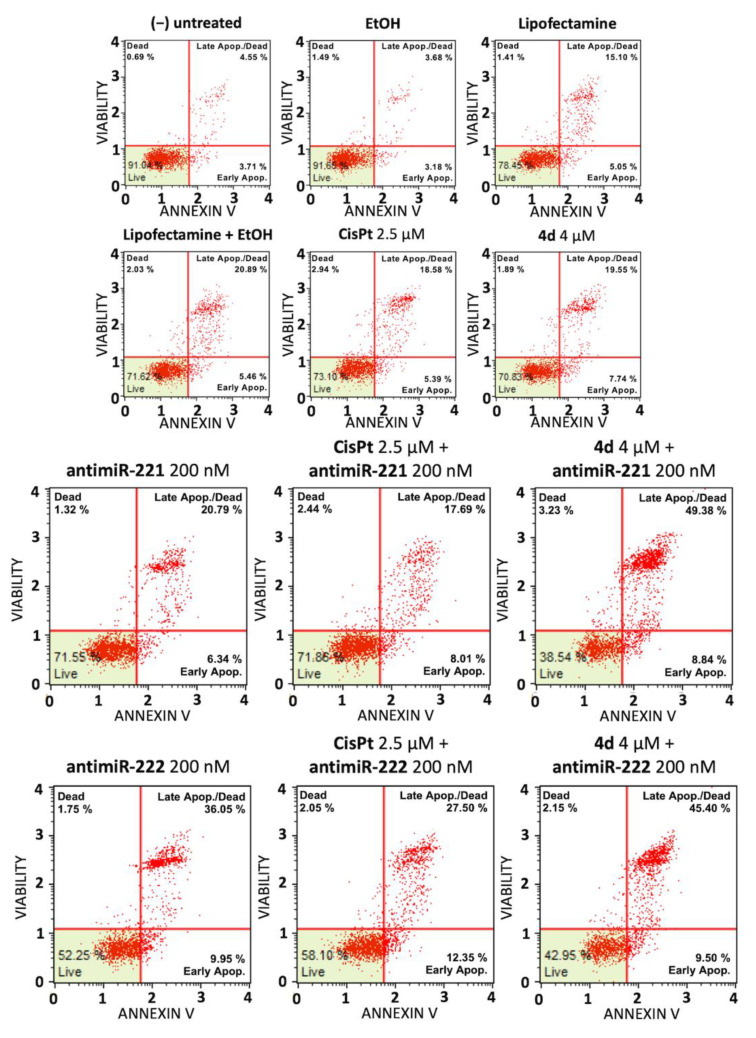
Induction of apoptosis of combined treatments based on the compound **4d** and inhibitors of miR-221-3p and miR-222-3p: glioblastoma T98G cell line. Annexin V assay was employed. T98G cells were treated for 48 h with the indicated experimental conditions, i.e., with the employed vectors or delivery systems (EtOH, lipofectamine, EtOH + lipofectamine) and with the singularly administered **4d** and the positive control CisPt (**upper** panel), with singularly administered miR-221-3p and miR-222-3p inhibitors and with the combined treatments using the miR-221-3p and miR-222-3p inhibitors and **4d** or CisPt (**lower** panel).

**Figure 8 pharmaceutics-15-01332-f008:**
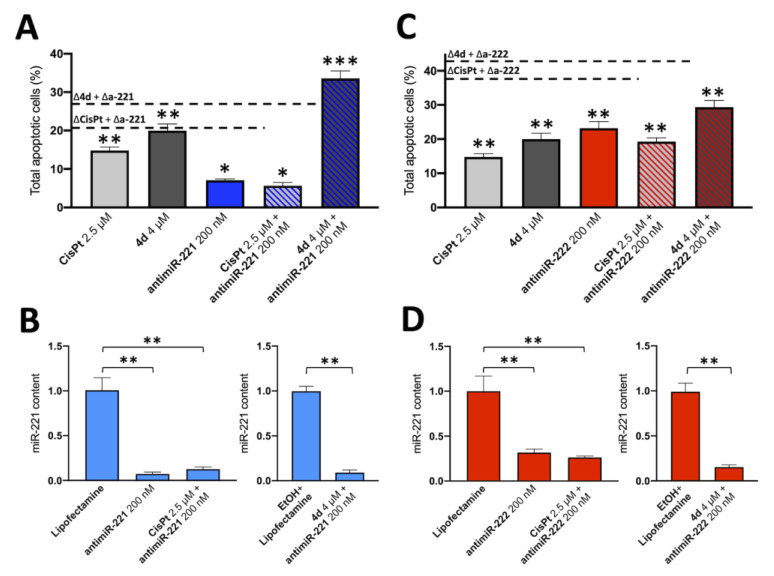
Summary of the experiments performed on the induction of apoptosis of combined treatments based on the compound **4d** and inhibitors of miR-221-3p and miR-222-3p: glioblastoma T98G cells. (**A**,**C**): Annexin V assay. (**B,D**): RT-qPCR analysis of the miR-221-3p and miR-222-3p cellular content. The dotted lines of panel (**A**,**C**) represent the sum of the singularly administered compounds, as indicated. Results represent mean ± SD (n = 3). *p* < 0.05 (*, significant), *p* < 0.01 (**; highly significant), *p* < 0.001 (***; highly significant).

**Figure 9 pharmaceutics-15-01332-f009:**
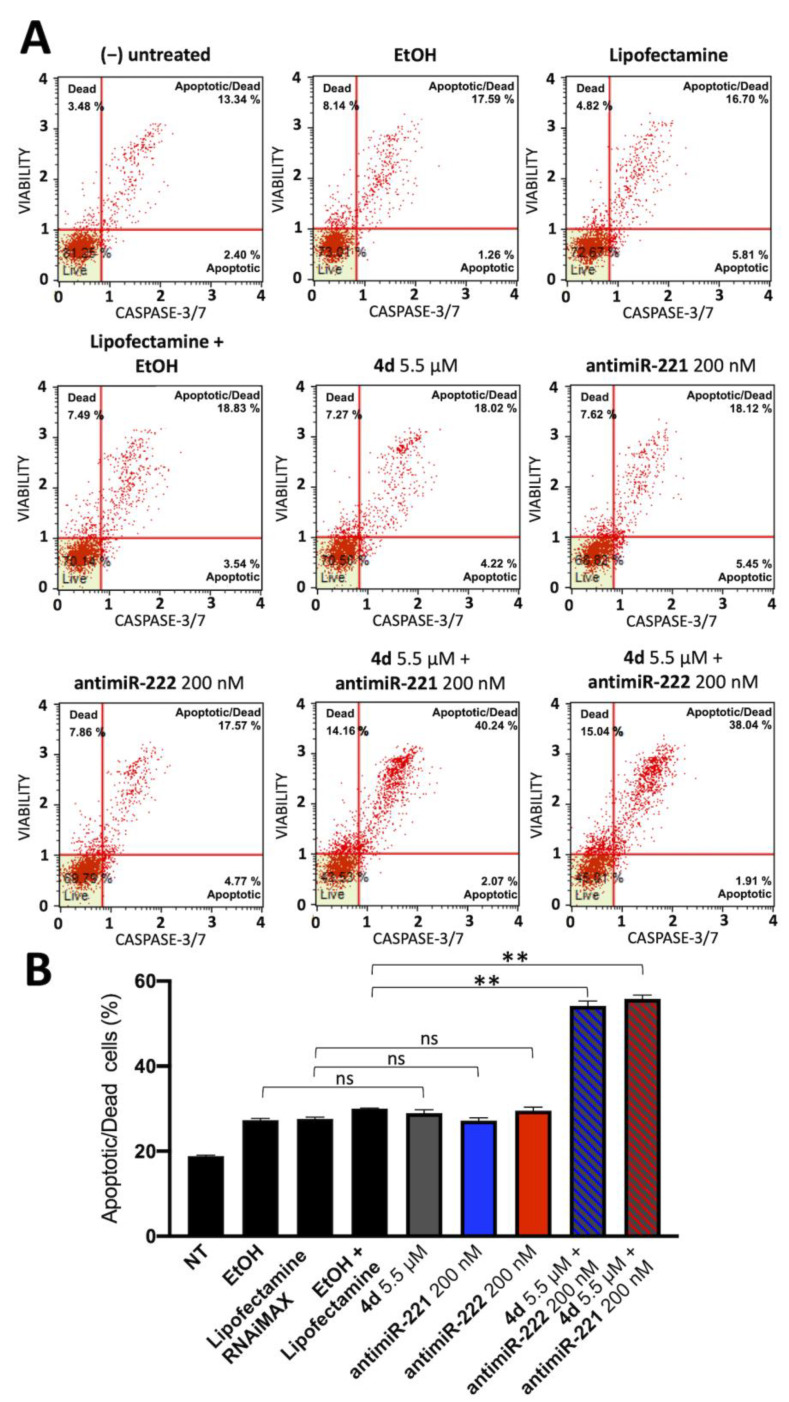
Summary of the experiments performed on U251 cell line showing caspase-3/7 activation following combined treatments based on the compound **4d** and inhibitors of miR-221-3p and miR-222-3p. (**A**): Representative dot plot obtained by flow cytometry following Caspase-3/7 staining. (**B**): Histogram representing the mean increase in apoptotic/dead cells, results represent mean ± SD (n = 3). *p* < 0.01 (**; highly significant), ns = not significant.

**Table 1 pharmaceutics-15-01332-t001:** List of assays employed for miRNA detection.

miRNA Name	Assay ID
hsa-miR-221-3p has-miR-222-3p	000524 000525
hsa-snRNA U6 hsa-let-7c-5p	001973 000379

## Data Availability

The datasets generated and/or analyzed during the present study are available from the corresponding author upon reasonable request.

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
