# Peer review of "Combined Treatment of Cancer Cells Using Allyl Palladium Complexes Bearing Purine-Based NHC Ligands and Molecules Targeting MicroRNAs miR-221-3p and miR-222-3p: Synergistic Effects on Apoptosis"

_pharmaceutics, 2023, doi:10.3390/pharmaceutics15051332_

Round 1

Reviewer 1 Report

The manuscript investigated the synergistic effect of the palladium allyl complex 4d and anti-miR-221/222-3p in triggering the cell apoptosis of distinct types of cancer cells. This synergistic effect in cancer cell death may lead to a promising therapeutic strategy for cancer treatment. Several points could help to improve the current manuscript.

-The statistics methods used in the manuscript should be described more clearly and when showing the statistical significance within the figure using the star, it’s better to indicate which groups are compared for each significance, especially for the quantification images with multiple groups.

-For Fig 1B, it’s better to provide the IC50 for each component in each cell type and the font size for axis labeling is too small here.

-For Fig 2, to improve the reproduction of the data, the quantification from 3 independent experiments is required. What is the staining reagent for y axis in the apoptosis assay in Fig 2, 4, 5, and 7? It’s better to label the staining reagent for y axis title.

-In the quantification results in Fig 6 and 8, why are the total apoptotic cells for controls missing? At least one control, either Etoh, Lipo, or Lipo+Etoh should be shown in the quantification results. Moreover, it’s meaningless to indicate the sum of singular treatment and the synergistic effect is indicated by the comparisons among combinations and singular treatment.

-One more critical problem: the representative images don’t indicate the average of the quantification in several images, such as 4d+antimir-222 in Fig 8C doesn’t match 4d+antimir-222 in Fig 7. The authors should check carefully in this point for other images.

-Several minor issues: 1) Typo errors: Fig 5 legend-“HT-29 cells” and page 3 line 121 “200nM antimir-221-3pp”. Please check carefully across the manuscript. 2) The citations 10-12 in the introduction section don’t match the text description. 3) Lots of highlighted texts in the manuscript. Are there any specific indications for these highlighted texts? 4) Fig S1 is missing.

Reviewer 2 Report

This manuscript report an extensive biological study of a Pd-allyl complex in combination with molecules targeting microRNAs.

The aim and the outcome of this work are very clear and the essays seem to be performed in a critical and rational way. The results are very promising and the research deserve to be deepen investigated.

From my point of the view the manuscript fits well on the journal aim and scope and the results are clear and well described/discussed. I would appreciate if the authors would explain a little bit more in detail the choice of these particular three complexes for this study and if they have any theory/hypothesis why compound 4d is the more active (role of the phosphine? role of the NHC ligand)?

What is the stability of these complexes in physiological conditions?

Reviewer 3 Report

The submitted study is aimed at a relatively attractive scientific area. The authors described biological studies of allyl derivatives of palladium on colon and glioblastoma cell lines. The added value of the work is the analysis of the mechanism of action of the most active allyl derivative of palladium in combination with antagomiRNA molecules (targeting two miRNAs: miR-221-3p and miR-222-3p). The experimental scheme includes only apoptosis assays. I am mainly missing verification of the tested combinations' effect on apoptosis activation proteins.

Here are my few comments and recommendations:

The IC50 must be written in as the IC50 in the whole manuscript.

Line 44 - please add a space

Line 65 - the authors conducted studies on HT-29, Ls174T and LoVo colorectal cancer cells or glioblastoma cell lines U251 and T98G. But in this sentence, they only write about one colorectal cancer cell line. Please rewrite/clarify.

Line 66 "Colorectal" - should be "colorectal" - lowercase

Line 66 - glioma cell lines U251 and T98G - change to glioblastoma cell lines

Lines 82-87 - please rephrase for clarity; please add more details about the paper results

Line 160 - what statistical test did the authors use in this study? Please add information to the experimental section.

Line 180 - "glioma" - refers to glioblastoma cell lines.

Figure 3 - please reword the figure caption

Please reword the results section. I feel that the figure caption, in many cases, is the same as in the text. The authors should be more specific about the results shown in the figures rather than „Figure 1B shows the effects of compounds 4d, 5d and 7a on cell proliferation of colon cancer (HT-29, Ls-174T and LoVo) and glioblastoma (U251 and T98G) cell lines. After 72 hours of treatment, cell cultures were analyzed to observe the antiproliferative effects of the allyl palladium complexes 4d, 5d and 7a, in comparison with the Cisplatin used at different concentrations.” The situation is similar throughout the section.

Throughout the manuscript, the authors should trace the nomenclature of the cell lines concerned. The T98G and U251 lines are glioblastoma lines, not glioma lines. The names glioma and glioblastoma are not synonymous. It appears as if the authors do not know what cells they are studying. In the introduction, the authors write about glioma, while in the discussion, they write about glioblastoma. Cell names are capitalized and lowercase once - please standardize the correct style throughout the manuscript.

Have the author's calculated combination ratios that give clear evidence of synergy?

The authors should verify the postulates made at the protein level by examining several proteins responsible for activating the apoptosis process (by the Western Blot technique).

Round 2

Reviewer 1 Report

One minor issue about the statistical methods: 

It's not suitable to use paired t-test across the manuscript. For most multiple comparisons among several groups, ANOVA test may be more suitable. 

Author Response

Dear Professor,

Reviewer of our article regarding a combined treatment of cancer cells using allyl palladium complexes bearing purine-based NHC ligands and molecules targeting microRNAs miR-221-3p and miR-222-3p, we thank you very much for your last suggestion (“It's not suitable to use paired t-test across the manuscript. For most multiple comparisons among several groups, ANOVA test may be more suitable”). We followed your advice, applying the ANOVA method (underlined in yellow in the manuscript, and, consequently, modifying some figures (figures 6, 8, in the main text, and S2-S7 in the Supplementary section), which were not, however, fortunately very different from those previously proposed.

Thank you for reading and correcting our manuscript.

Best regards

Ilaria Lampronti

Reviewer 3 Report

The authors have addressed all my comments. The manuscript in its present form is suitable for publication.

Author Response

Dear Professor,

Reviewer of our article regarding a combined treatment of cancer cells using allyl palladium complexes bearing purine-based NHC ligands and molecules targeting microRNAs miR-221-3p and miR-222-3p, we thank you very much for reading and correcting our manuscript.

Best regards

Ilaria Lampronti
